# Blood Lead (Pb) Levels as a Possible Marker of Cancer Risk in a Prospective Cohort of Women with Non-Occupational Exposure

**DOI:** 10.3390/biomedicines13071587

**Published:** 2025-06-29

**Authors:** Krzysztof Lubiński, Marcin R. Lener, Wojciech Marciniak, Jakub Pawłowski, Julia Sadzikowska, Adam Kiljańczyk, Milena Matuszczak, Piotr Baszuk, Sandra Pietrzak, Róża Derkacz, Marta Bryśkiewicz, Cezary Cybulski, Jacek Gronwald, Tadeusz Dębniak, Tomasz Huzarski, Steven A. Narod, Rodney J. Scott, Jan Lubiński

**Affiliations:** 1International Hereditary Cancer Center, Department of Genetics and Pathology, Pomeranian Medical University in Szczecin, ul. Unii Lubelskiej 1, 71-252 Szczecin, Poland; krzychul123456789@gmail.com (K.L.); marcinlener@poczta.onet.pl (M.R.L.); wojciech.marciniak@read-gene.com (W.M.); pawlowski.jakub@icloud.com (J.P.); julia.sadzikowska26@gmail.com (J.S.); adam.kiljanczyk@pum.edu.pl (A.K.); milena.matuszczak@pum.edu.pl (M.M.); piotr.baszuk@pum.edu.pl (P.B.); sandra.pietrzak@pum.edu.pl (S.P.); roza.derkacz@gmail.com (R.D.); marta.bryskiewicz@pum.edu.pl (M.B.); cezarycy@pum.edu.pl (C.C.); jgron@pum.edu.pl (J.G.); debniak@pum.edu.pl (T.D.); huzarski@pum.edu.pl (T.H.); 2Read-Gene, ul. Alabastrowa 8, 72-003 Grzepnica, Poland; 3Department of Diagnostic Imaging and Interventional Radiology, Pomeranian Medical University Hospital No 1, 71-252 Szczecin, Poland; 4Department of Clinical Genetics and Pathology, University of Zielona Góra, ul. Zyty 28, 65-046 Zielona Góra, Poland; 5Women’s College Research Institute, Toronto, ON M5S 1B2, Canada; steven.narod@wchospital.ca; 6Dalla Lana School of Public Health, University of Toronto, Toronto, ON M5T 3M7, Canada; 7School of Biomedical Sciences and Pharmacy, Centre for Information-Based Medicine, Hunter Medical Research Institute, University of Newcastle, Newcastle, NSW 2305, Australia; rodney.scott@newcastle.edu.au; 8Division of Molecular Medicine, Pathology North, NSW Pathology, Newcastle, NSW 2305, Australia

**Keywords:** Pb, cancer risk, prospective study, non-occupational exposure, carcinogen

## Abstract

**Background/Objectives**: To correlate blood lead (Pb) levels with cancer risk in a prospective cohort of healthy women with non-occupational exposure to lead. We hypothesize that blood Pb levels can predict the risk of cancer in healthy women. **Methods**: The study was performed with women registered at the Hereditary Cancer Centre, Szczecin, aged 40 years and above between September 2010 and March 2024. A total of 2927 unaffected women were included in the study. Exclusion criteria were BRCA1 gene mutation, women with diagnosed cancer, and women with occupational exposures to Pb. All patients were asked about their occupational exposure and tested for the three Polish BRCA1 founder mutations (c.5266dupC/5382insC; c.181T > G/300T > G; c.4035delA/4153delA). Inductively coupled plasma mass spectrometry was used to measure blood Pb levels. The study was blinded to all scientists involved, and all samples were assayed in the absence of any knowledge about the clinical status of each participant. **Results**: There were 239 incident cancers diagnosed in the cohort after an average follow-up of 6 years. Compared to women with the lowest blood Pb concentration, women with higher blood Pb levels had a significantly increased risk of developing any cancer (HR = 1.46; (95% CI: 1.006–2.13; *p* = 0.046)). The association was stronger for women below the age of 50 years at study entry (HR = 2.59; (95% CI: 1.37–4.89; *p* = 0.003)). For women over 50 years of age, the results were statistically insignificant. **Conclusions**: This study suggests that blood Pb levels have the potential to be used as a marker of cancer risk in women under 50 years of age who have no known occupational exposure to this heavy metal. Further investigations using additional groups of women from Poland and other countries are needed for validate these findings.

## 1. Introduction

Heavy metals comprises a group of metal and metalloid elements that have been linked to contamination and potential toxicity. Over the years many definitions of heavy metals have emerged that include definitions in terms of density (specific gravity), atomic weight (relative atomic mass), atomic number, and other chemical properties associated with toxicity. Although there is no consensus on the definition, the term “heavy metal” is commonly used to refer to metals that have a harmful effect on animal and human health [1]. According to the IARC, heavy metals are defined as carcinogenic or probably carcinogenic to humans and include arsenic, beryllium, cadmium, chromium (Vl) compounds, nickel, and lead (Pb). Inorganic lead is classified as probably carcinogenic to humans (listed as a group 2A carcinogen), and at present, organic lead is classified as group 3 (unclassifiable) [2].

The extensive use of Pb has resulted in widespread environmental contamination, leading to significant public health concerns [3,4,5]. Pb contamination of the environment has primarily occurred via its use as a fuel additive (tetraethyl Pb-TEL), which significantly increased atmospheric Pb levels. The use of TEL was estimated to account for between 80 and 90% of atmospheric Pb in major cities. As a consequence, it was also the main source of exposure to the human population. Since the ban of TEL in petrol, there has been a significant population-wide drop in Pb exposure, which is reflected in an overall decrease in blood Pb levels [6].

At present, important sources of environmental Pb contamination include mining, smelting, manufacturing and recycling [3,4,7,8]. Much of the Pb in global commerce is now obtained by recycling [9,10]. Vegetables, cereals, fruit, and food products are also potential sources of Pb, where inorganic Pb is absorbed from industrial dust or engine exhaust fumes [4,11]. Due to water pollution, high levels of Pb are also present in some pelagic fish species [12]. The most striking is in tuna, which can accumulate high levels of heavy metals through predation, making it a good biomonitor of water pollution. Waterways with heavy traffic of oil tankers are associated with the risk of elevated blood Pb levels in fish that inhabit those regions [12]. Pb paint hazards were noted in households built before 1978 when Pb paint was not generally prohibited. Other sources of Pb exposure are drinking water service lines made from Pb, Pb solder, or plumbing materials that contain Pb [13].

Most of the exposure of adults to Pb occurs in the workplace. Exposure to Pb from non-occupational sources is also widespread in the adult population; however, the levels are generally lower than those found in the workplace. People of all ages are exposed to Pb through contact with Pb in the air, dust, soil, and drinking water. The main routes of ingestion are inhalation of particulate matter containing Pb and swallowing of contaminated food. Skeletal Pb is the largest reserve of this heavy metal. Approximately 99% of all Pb in the human body is found in bone, and the release of Pb from bone stores is the cause of considerable inter-individual variability. Pb’s toxic effects occur mainly in three systems: erythrocytes and their precursors, the central and peripheral nervous system, and the kidneys. Pb causes hematological problems such as anemia. Pb is a neurotoxin for both the peripheral and central nervous system, causing intellectual disability, shortened attention span, slower reaction times, and extensor muscle palsy with wrist and ankle drop. Chronic nephropathy, which may progress to kidney failure, is the most common Pb poisoning nephrotoxicity event. An increased incidence of hypertension and cerebrovascular accidents has been reported with long-term, high-dose exposure to Pb. Reproductive toxicity in workers of both sexes exposed to high doses of Pb has been described, where the incidence of spontaneous miscarriage has been reported to be higher in female workers exposed to Pb. Male workers exposed to Pb have reduced sperm counts [14,15,16,17,18].

Under conditions of bone demineralization, such as during pregnancy and lactation, Pb can be released into the blood. After menopause, there is a significant increase in whole blood and plasma Pb levels. Pb may also interact with other factors in the course of postmenopausal osteoporosis, worsening disease progression. Pb is known to inhibit the activation of vitamin D and the absorption of calcium from food and a number of regulatory aspects of the function of osteocytes [19,20,21].

Pb diminishes the activity of various enzymes by substituting calcium, resulting in changes in the structure and activity of the respective proteins. Examples include enzymes involved in the addition of sulfhydryl and amide [22,23,24,25].

Possible mechanisms of Pb carcinogenic activity include direct DNA damage, clastogenicity, or inhibition of DNA synthesis or repair. Pb may also generate reactive oxygen species that alter DNA and protein structures. Recent data indicate that Pb can also serve as a substitute for zinc and alter the ability of zinc finger binding, resulting in aberrant transcription [26,27,28].

In a meta-analysis released in 2024 [28], 11 studies investigated cancer risk in relation to human blood Pb levels. Most of these studies are population based, and two focused on occupational exposure. Occupational exposure showed that elevated blood Pb levels resulted in a statistically significant increased risk of esophageal and lung cancers. Population studies showed that elevated blood Pb levels resulted a significant increased risk of any cancer including gastroesophageal cancers. A populational study of blood Pb levels and breast cancer risk suggested that there was little evidence linking blood Pb levels with breast cancer risk. The meta-analysis was not specifically designed to determine breast cancer risk since it was a population study aiming to obtain a consensus on cancer risk in general (Appendix A).

The current study was undertaken to better define the relationship between Pb exposure and cancer risk. To address the shortfall in knowledge about the relationship between Pb and cancer risk, especially in Poland, we undertook a study using a large cohort of unaffected women where we measured baseline Pb levels and documented incident cancers with an average of 6 years of follow-up. We included the blood arsenic level in multivariate analysis due to the fact that blood arsenic (As) level has been recognized recently as an additional significant marker of cancer risk in women [29].

## 2. Materials and Methods

### 2.1. Study Group

The study was conducted among women aged 40 years and above who had undergone genetic testing for at least three founder mutations most frequently observed in Poland (c.5266dupC–5382insC; c.181T > G-300T > G; c.4035delA–4153delA) at the Pomeranian Medical University in Szczecin between September 2010 and March 2024. These three founder mutations account for over 90% of all BRCA1 dependent inherited predispositions to breast cancer in Poland [30]. At the time of enrollment, none of the participants had been diagnosed with cancer. All subjects in the study gave their written informed consent to take part in the study and agreed to provide a blood sample for research purposes. During the follow-up period, patients were contacted at annual visits for routine monitoring. The study was conducted in accordance with the Helsinki Declaration and with the consent of the Ethics Committee of Pomeranian Medical University in Szczecin (number KB-0012/73/10 of 21 June 2010). Informed patient consent was obtained. A blood sample for genetic testing was taken at the first outpatient visit. For research purposes, a separate 10 mL aliquot of whole blood was also collected and stored at a temperature of −80 °C. From Monday to Friday, blood samples were taken between 8 am and 2 pm. Subjects were asked to complete a detailed questionnaire that included family history of cancer, age, smoking status, hormone use, and personal medical history, including oophorectomy. The exclusion criteria were BRCA1 mutation carriers tested in all participants of the study (c.5266dupC–5382insC; c.181T > G-300T > G; c.4035delA–4153delA), women with any diagnosed cancer, aged under 40 years of age, and women with occupational exposure to Pb.

### 2.2. Measurement of Blood Pb

Briefly, 10 mL of peripheral blood was collected into a vacutainer tube containing sodium ethylenediaminetetraacetic acid (EDTA) from all study participants and then stored at −80 °C until the day of analysis. Study subjects were fasting for at least six hours before sample collection. The mean time that elapsed since the date of blood collection and the date of Pb measurement was 41 months (range 6–105 months).

On the day before analytical procedures, samples were thawed at room temperature. All samples were gently vortexed before taking an aliquot for determination of metals. The determination of ^208^Pb and ^75^As was carried out using the ICP-MS mass spectrometer ELAN DRC-e (PerkinElmer, Concord, Toronto, ON, Canada). Oxygen was used as a reaction gas. Arsenic was measured in mass-shift mode at mass 91 as ^91^AsO^+^. The ICP-MS mass spectrometer was tuned to the manufacturer’s criteria prior to analysis. For maximum sensitivity, critical parameters have been optimized. An external calibration technique was used to calibrate the spectrometer. Calibration standards were freshly prepared daily using 10 µg/mL Multi-Element Calibration Standard 3 (PerkinElmer Pure Plus, Shelton, CT, USA), diluting it with a blank reagent to the final levels of 0.5 µg/L, 1 µg/L, 2 µg/L, 5 µg/L, and 10 µg/L for Pb level determination and 0.2 µg/L, 0.5 µg/L, 0.75 µg/L, and 1.00 µg/L for As determination. Correlations for calibration curves were always higher than 0.999. The calibration was performed using matrix matching. The internal standard was rhodium (^103^Rh). Blood was diluted 40 times in a blank reagent (70 µL of blood + 2730 µL of blank reagent). The blank reagent contained high purity water (>18 MΩ), TMAH (AlfaAesar, Kandel, Germany), Triton X-100 (PerkinElmer, Shelton, CT, USA), ethanol (Merck, Darmstadt, Germany), rhodium (PerkinElmer, Shelton, CT, USA), and EDTA (Sigma-Aldrich, Leuven, Belgium).

Due to the specificity of the measurement, a tetramethylammonium hydroxide solution was used for dilutions. The alkaline pH ensured good solubility of blood components, without causing precipitation of any fraction. Additionally, for better dispersion of dissolved blood components, an addition of a non-ionic surfactant in the form of Triton T-100 was used. This compound not only facilitates the dissolution of proteins, among others, but also contributes to faster rinsing of the sample from the spectrometer introduction system. The addition of edetic acid was used to obtain the stability of metal ions dissolved in the solution. Additionally, due to the high content of carbon-containing compounds, ethanol was added to all solutions to eliminate the effect associated with a significant amount of carbon in the tested sample.

The following LOD and LOQ values were obtained 0.137 µg/L and 0.459 µg/L for lead and 0.0328 µg/L and 0.1094 for arsenic, respectively.

#### Quality Control

The accuracy and precision of all measurements were tested using certified reference material CRM Clincheck Plasmonorm Blood Trace Elements Level 1 (Recipe, Munich, Germany). Recovery rates were between 80 and 105% for analyzed elements; calculated recurrency (Cv%) was below 15% for lead. The testing laboratory is a member of two independent external quality assessment schemes: the LAMP organized by the CDC (LAMP: Pb And Multielement Proficiency Program; CDC: Centers for Disease Control) and the QMEQAS organized by the Institute National de Santé Publique du Québec (QMEQAS: Quebec Multielement External Quality Assessment Scheme). Batch effect was controlled by measuring CRM every 90 samples. In addition, an in-lab prepared quality control sample was measured in the same manner (QCS2 mean-11.6; SD 0.61; RSD 5.29).

### 2.3. Statistical Analysis

Information about cancer incidence was obtained from the medical records and the pathology reports of the treating hospitals. The subjects were divided into quartiles according to the blood Pb levels in the unaffected women, ensuring the same number of women in each quartile. Study subjects were followed from the date of blood collection or age 40 (whichever came last) until the first cancer or death from another cause until August 2024 or the date of last contact. The differences in cancer risk according to Pb quartile were evaluated on the basis of hazard ratios generated using multivariate Cox proportional hazard models: adjusted by age at blood draw (<50 and ≥50), smoking, cancers in first degree relatives, oophorectomy, oral contraception, hormone replacement therapy, and serum arsenic levels (quartiles). One of additional features in the multivariate analysis is arsenic. It was reported in 2020 that arsenic is a strong breast cancer risk feature in women [30].

## 3. Results

The study cohort consisted of 2927 cancer-free women. Blood Pb levels were measured in a single sample from each woman. The average subject age at the time of blood collection was 53 years (range 40–84 years). The characteristics of individuals in the study are presented in Table 1. The mean blood Pb level was 14.49 µg/L. The mean blood Pb levels for various subgroups are presented in Table 1. Blood Pb levels were higher for women at age ≥50 compared to women under age 50 (*p* < 0.01) and were also higher for current smokers compared to non-smokers (*p* < 0.01). The distribution of the blood Pb levels is shown in Figure 1.

From the date of blood collection, the women were followed for a mean time of 6 years and 2 months (range 0.5 to 13.5 years). Throughout the follow-up period, 239 cancer incidents occurred, of which 116 cases were breast cancer and 123 cases of cancer at other sites (Table 2). We also performed an analysis of the entire group of women (Table 3 and Table 4) and for the groups <50 and ≥50 years (Appendix A). The ten year cumulative cancer risk by blood Pb quartiles is shown in Figure 2.

The 2927 women were divided into four categories (quartiles) of similar size of unaffected females according to their total blood Pb level. The cut-off levels to define the quartiles were (Q1) 2.58–9.39 µg/L, (Q2) 9.4–12.58 µg/L, (Q3) 12.59–17.17 µg/L, and (Q4) 17.18–96.27 µg/L. The hazard ratios (HRs) for developing any cancer according to blood Pb level quartile are presented in Table 3. A blood level greater than 9.39 µg/L was associated with an increased risk of developing any cancer. The hazard ratio for women in quartiles Q2–Q4 combined compared to quartile 1 was 1.46 (95% CI: 1.006–2.13; *p* = 0.046). Among the subgroup of women below 50 years of age, the observed effect was stronger for any cancer (Q1 vs. Q2–4; HR = 2.59 (95% CI: 1.37–4.89; *p* = 0.003) and breast cancer only (Q1 vs. Q2–4; HR = 2.64; (1.21–5.76; *p* = 0.014) (Table 5 and Table 6).

## 4. Discussion

A statistically significant association between blood Pb levels and an increased risk of cancer in women was observed. For women under the age of 50 years, there was a greater risk of cancer with higher Pb levels compared to the entire group (HR = 2.59 (95% CI: 1.37–4.89; *p* = 0.003)). These results reveal that high Pb blood levels are an unfavorable predictor of cancer risk in premenopausal women. For postmenopausal women, there appeared to be no association with cancer risk and blood Pb levels. However, women over 50 years of age generally had higher Pb levels compared to women of younger ages, suggesting the sequestering of Pb that is stored over longer periods of time may no longer be as reactive compared to that more recently ingested.

By comparing the results of this study with those already published, it appears that Pb is more likely to significantly increase the risk of cancer in women under 50 years of age compared to older women. It is notable that in young women, serum estrogen levels correlate with environmental exposure to lead [31]. The use of estrogen-based contraception increases the risk of breast cancer. This supports the hypothesis that the increased risk of cancer in young women exposed to lead may manifest through modifying estrogen exposures [32,33]. This is further supported by the high levels of estradiol in in young women. In contrast, in older women, estradiol is replaced by estrone, which has not been observed to interact with Pb [34].

Another modifier of Pb’s effect on cancer risk may be related to DNA variants involved in Pb metabolism; however, any difference in protection against cancer would be expected to affect all women and not just those under 50 years of age. One aspect that we have not addressed is the recency of exposure. If the sequestration of Pb in bone results in it becoming less bioavailable to other tissues, then the risk of cancer might diminish with time since last exposure. This aspect was not the focus of this report and awaits further investigation to better understand why young women appear more at risk of malignancy when exposed to Pb.

Herein, we have shown in a prospective cohort of women that Pb is a potential cancer risk factor and that Pb is significantly associated with breast cancer risk in young women. Blood samples collected prior to 2010 were all found to have lead levels higher than those seen in the lowest quartile in the current study (our reference group), and studies in this era would not be expected therefore to show the association with low Pb levels. Our samples were all collected after 2010. After this time, environmental exposure to PB was subject to several control measures in Poland.

Given that young women with higher blood Pb levels had an almost 3-fold increased risk of developing breast cancer compared to women with the lowest levels, the cut-off values of 9.39 µg/L should be further explored.

### Limitations

Whole blood has historically been the reference matrix for assessing Pb exposure (>99% of total blood Pb is bound to red blood cells) [35]. The blood Pb half-life is 4 to 5 weeks, reflecting recent exposure and Pb mobilization from other tissues including bone [36]. Thus, a single blood Pb measurement cannot distinguish between recent exposure and previous exposures due to the cyclic movement of Pb between different tissue compartments (i.e., blood and bone/soft tissue) [37,38].

Females included in the study have been asked about occupational exposure; however, their socio-economic status was not checked. A potential additional limitation is that blood Pb level was measured only from one sample at recruitment. Another limitation is that this study was not performed in different regions/ethnic groups. The results will potentially be more specific if information on DNA variants involved Pb metabolism are added. In addition, with the exception of BRCA1, the status of other high risk genes is not known.

## 5. Conclusions

This study demonstrates that blood Pb levels have the potential to be used to aid in the assessment of cancer risk in younger women (<50 years of age) and women without occupational exposure. Further investigations using additional groups of women from Poland and other countries are required to validate these findings. It is important that this cohort be expanded and followed for a longer period to better assess the relationship between blood Pb levels and cancer risk.

## Figures and Tables

**Figure 1 biomedicines-13-01587-f001:**
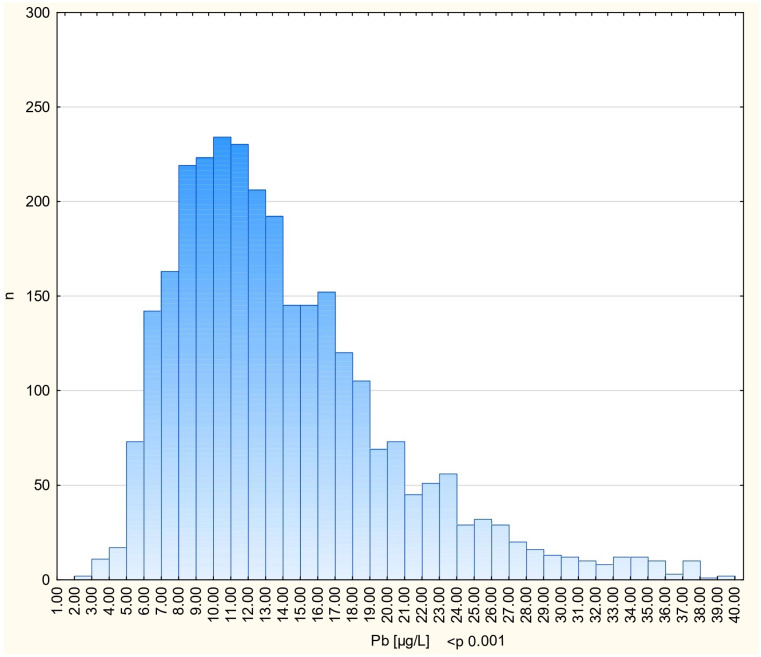
The distribution of blood Pb levels in all patients.

**Figure 2 biomedicines-13-01587-f002:**
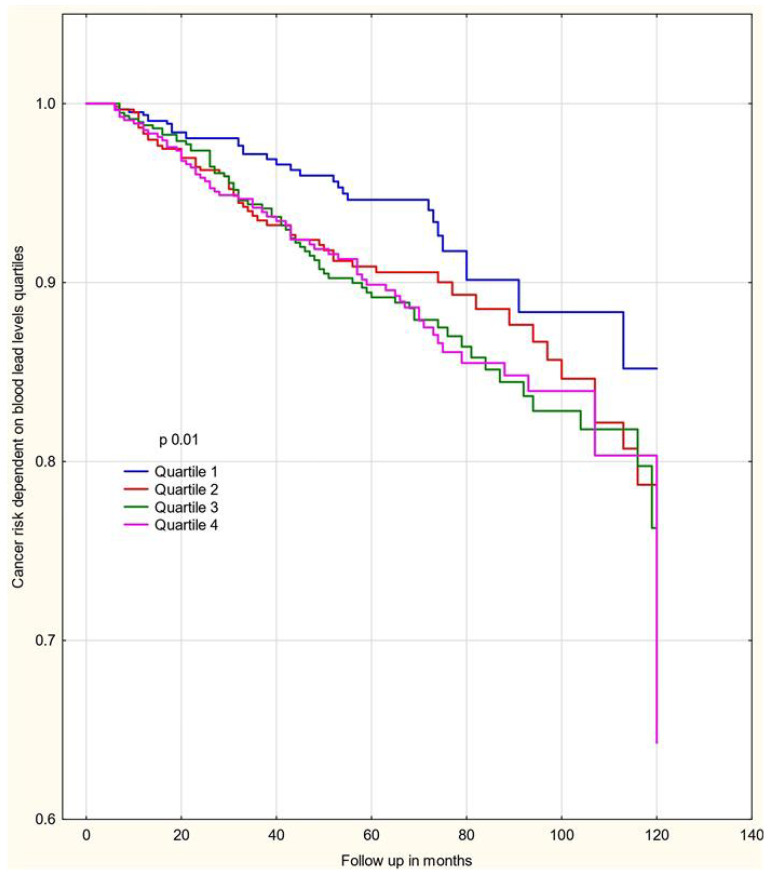
Ten-year cumulative cancer risk as a function of blood Pb levels.

**Table 1 biomedicines-13-01587-t001:** Characteristics of 2927 women in the cohort.

Characteristics	Unaffected	Cases	Mean Pb Level 14.49 µg/L, SD 7.79	Univariate Cancer Risk HR; (95% CI); *p*	Multivariate Cancer Risk HR; (95% CI); *p*
**Blood Pb levels by quartiles** µg/L					
Q1 2.58–9.39	672 (22.95%)	33 (1.2%)	7.48 ± 1.34		
Q2 9.4–12.58	672 (22.95%)	60 (2.14%)	10.95 ± 0.90	HR 1.52; 95% CI (0.99–2.33); *p* = 0.05	HR 1.48; 95% CI (1.06–2.08); *p* = 0.02
Q3 12.59–17.17	672 (22.95%)	73 (2.495%)	14.75 ± 1.39	HR 1.66; 95% CI (1.1–2.5); *p* = 0.015	HR 1.57; 95% CI (1.03–2.39); *p* = 0.03
Q4 17.18–96.27	677 (22.95%)	68 (2.32%)	24.35 ± 8.84	HR 1.48; 95% CI (0.97–2.24); *p* = 0.06	HR 1.34; 95% CI (0.87–2.06); *p* = 0.18
**Blood arsenic levels by quartiles** µg/L					
Q1 <0.599	688 (23.5%)	40 (1.37%)	13.66 ± 7.84		
Q2 0.60–0.81	676 (23.09%)	51 (1.7%)	13.85 ± 6.80	HR 1.23; 95% CI (0.81–1.86); *p* = 0.31	HR 1.20; 95% CI (0.79–1.82); *p* = 0.38
Q3 0.81–1.24	668 (22.8%)	70 (2.4%)	15.13 ± 7.47	HR 1.71; 95% CI (1.16–2.53); *p* = 0.006	HR 1.68; 95% CI (1.14–2.49); *p* = 0.008
Q4 >1.243	661 (22.6%)	73 (2.495%)	15.31 ± 8.79	HR 1.89; 95% CI (1.28- 2.78); *p* = 0.001	HR 1.85; 95% CI (1.26–2.74); *p* = 0.001
**Cancers in first degree relatives**					
No	463 (15.8%)	37 (1.3%)	14.10 ± 7.78		
Yes	2230 (76.2%)	197 (6.7%)	14.57 ± 7.79	HR 1.11; 95% CI (0.80–1.62); *p* = 0.46	HR 1.11; 95% CI (0.78–1.59); *p* = 0.53
**Oral contraceptives**					
No	1983 (67.75%)	188 (6.41%)	15.07 ± 7.74		
Yes	710 (24.26%)	46 (1.58%)	12.83 ± 7.72	HR 0.86; 95% CI (0.62–1.19); *p* = 0.38	HR 0.89; 95% CI (0.64–1.25); *p* = 0.52
**Oophorectomy**					
No	2527 (86.33%)	216 (7.37%)	14.38 ± 7.73		
Yes	166 (5.67%)	18 (0.63%)	16.20 ± 8.49	HR 1.26; 95% CI (0.78–2.05); *p* = 0.33	HR 1.17; 95% CI (0.72–1.92); *p* = 0.51
**Hormone replacement therapy**					
No	2127 (72.66%)	176 (6.02%)	14.24 ± 7.63		
Yes	566 (19.33%)	58 (1.98%)	15.45 ± 8.29	HR 1.21; 95% CI (0.9–1.63); *p* = 0.2	HR 1.11; 95% CI (0.82–1.51); *p* = 0.49
**Smoking status**					
No	1392 (47.55%)	119 (4.065%)	13.45 ± 7.06		
Yes	1301 (44.44%)	115 (3.945%)	15.61 ± 8.36	HR 0.98; 95% CI (0.76–1.27); *p* = 0.9	HR 0.96; 95% CI (0.74–1.25); *p* = 0.78
**Age**					
<50	1154 (39.42%)	63 (2.15%)	11.69 ± 6.20		
≥50	1539 (52.58%)	171 (5.85%)	16.49 ± 8.18	HR 1.61; 95% CI (1.21–2.16); *p* = 0.001	HR 1.49; 95% CI (0.771.38); *p* = 0.019

**Table 2 biomedicines-13-01587-t002:** Incident cancers detected in the cohort.

Cancer Site	n	Cases (%)	Mean Pb Level µg/L, SD
None	2693		14.34 ± 7.60
Any cancer	239	100	16.28 ± 9.56
Breast	116	48.5	16.69 ± 10.79
Lung	11	4.6	22.50 ± 10.65
Uterus	14	5.8	17.65 ± 11.06
Leukemia	4	1.7	17.90 ± 11.27
Lymphoma	5	2	17.08 ± 1.37
Bladder	4	1.7	17.07 ± 5.32
Thyroid	12	5	16.75 ± 7.88
Ovarian	15	6.3	15.91 ± 8.35
Cervix	7	2.9	15.74 ± 9.18
Myeloma	3	1.2	15.17 ± 5.41
Melanoma	8	3.3	14.74 ± 6.80
Liver	1	0.5	14.21
Stomach	5	2	13.77 ± 6.73
Skin	9	3.8	13.59 ± 5.43
Glioma	1	0.5	13.50
Chondroma	1	0.5	13.35
Colon	14	5.8	13.31 ± 7.90
Parotid gland	1	0.5	12.56
Kidney	5	2	12.04 ± 3.69
Abdominal cavity	1	0.5	11.11
Pancreas	2	0.9	10.6 ± 2.81

**Table 3 biomedicines-13-01587-t003:** Hazard ratios for any cancer by blood Pb level for the whole group (quartiles).

			Univariate COX Regression			Multivariate COX Regression *		
Blood Pb Level µg/L	Cases	Unaffected	HR	95% CI	*p*-Value	HR	95% CI	*p*-Value
Q1 2.58–9.39	33 (4.7%)	672 (95.3%)	—	—	—	—	—	—
Q2 9.4–12.58	60 (8.2%)	672 (91.8%)	1.52	0.99–2.33	0.05	1.48	0.96–2.27	0.07
Q3 12.59–17.17	73 (9.8%)	672 (90.2%)	1.66	1.1–2.5	0.015	1.57	1.03–2.38	0.03
Q4 17.18–96.27	68 (8.8%)	677 (91.2%)	1.48	0.97–2.24	0.06	1.34	0.87–2.06	0.18
Q1 2.58–9.39 vs. Q2–Q4 9.4–96.27	201 (9%)	2021 (91%)	1.55	1.07–2.25	0.018	1.46	1.006–2.13	0.046

* Adjusted for smoking, first degree relatives, adnexectomy, oral contraception, hormone replacement therapy, and arsenic quartiles levels.

**Table 4 biomedicines-13-01587-t004:** Hazard ratios for breast cancer by blood Pb level for the whole group (quartiles).

			Univariate COX Regression			Multivariate COX Regression *		
Pb Level µg/L	Cases	Unaffected	HR	95% CI	*p*-Value	HR	95% CI	*p*-Value
Q1 2.58–9.39	17 (2.46)	672 (97.54%)	—	—	—	—	—	—
Q2 9.4–12.58	31 (4.4%)	672 (95.6%)	1.55	0.85–2.80	0.14	1.53	0.84–2.77	0.16
Q3 12.59–17.17	31 (4.4%)	672 (95.6%)	1.40	0.77–2.53	0.26	1.37	0.74–2.50	0.30
Q4 17.18–96.27	37 (5.1%)	677 (94.9%)	1.58	0.88- 2.82	0.11	1.42	0.78–2.59	0.24
Q1 2.58–9.39 vs. Q2–Q4 9.4–96.27	99 (4.66%)	2021 (95.34%)	1.51	0.90–2.53	0.11	1.44	0.85–2.44	0.17

* Adjusted for smoking, first degree relatives, adnexectomy, oral contraception, hormone replacement therapy, and arsenic quartiles levels.

**Table 5 biomedicines-13-01587-t005:** Hazard ratios for any cancer by blood Pb level for the <50-year age group (quartiles).

			Univariate COX Regression			Multivariate COX Regression *		
Pb Level µg/L	Cases	Unaffected	HR	95% CI	*p*-Value	HR	95% CI	*p*-Value
Q1 2.58–9.39	12 (2.3%)	492 (97.7%)	—	—	—	—	—	—
Q2 9.4–12.58	23 (6.6%)	322 (93.4%)	2.5	1.24–5.03	0.01	2.49	1.23–5.04	0.01
Q3 12.59–17.17	19 (8.5%)	204 (91.5%)	2.9	1.41–6.04	0.003	3.04	1.46–6.30	0.002
Q4 17.18–96.27	9 (6.2%)	136 (93.8%)	2.1	0.89–5.05	0.08	2.12	0.88–5.11	0.09
Q1 2.58–9.39 vs. Q2–Q4 9.4–96.27	51 (7.15%)	662 (92.85%)	2.56	1.36–4.81	0.003	2.59	1.37–4.89	0.003

* Adjusted for smoking, first degree relatives, adnexectomy, oral contraception, hormone replacement therapy, and arsenic quartiles levels.

**Table 6 biomedicines-13-01587-t006:** Hazard ratios for breast cancer by blood Pb level for the <50-year age group (quartiles).

			Univariate COX Regression			Multivariate COX Regression *		
Pb Level µg/L	Cases	Unaffected	HR	95% CI	*p*-Value	HR	95% CI	*p*-Value
Q1 2.58–9.39	8 (1.6%)	492 (98.4%)	—	—	—	—	—	—
Q2 9.4–12.58	19 (5.5%)	322 (94.5%)	3.08	1.34–7.05	0.007	3.20	1.39–7.359	0.006
Q3 12.59–17.17	9 (4.2%)	204 (95.8%)	2.13	0.82–5.55	0.11	2.23	0.85–5.85	0.10
Q4 17.18–96.27	6 (4.2%)	136 (95.8%)	2.10	0.72–6.10	0.16	2.018	0.688–5.92	0.20
Q1 2.58–9.39 vs. Q2–Q4 9.4–96.27	34 (4.8%)	662 (95.2%)	2.57	1.18–5.57	0.016	2.64	1.21–5.76	0.014

* Adjusted for smoking, first degree relatives, adnexectomy, oral contraception, hormone replacement therapy, and arsenic quartiles levels.

## Data Availability

Data supporting the results presented are available from the authors upon request from any interested researchers.

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
