# Peer review of "Blood Lead (Pb) Levels as a Possible Marker of Cancer Risk in a Prospective Cohort of Women with Non-Occupational Exposure"

_biomedicines, 2025, doi:10.3390/biomedicines13071587_

Round 1
Reviewer 1 Report (Previous Reviewer 2)
Comments and Suggestions for Authors
The article needs to be improved and further clarifications are needed before publication.
Material and Methods
What is the study design? It appears to be an analytical observational study, more precisely a cohort design. Follow the STROBE guidelines (https://www.strobe-statement.org/).
Inclusion and exclusion criteria must be clearly defined.
Lanes 134 and 148, replace “ml” with “mL”. Use SI correctly. Check it through the ms.
Results
The characterization of the study population is not evident. The first outcome in an analytical study is to describe the study population.
Add the flowchart showing the women enrolled in the observational study and who were excluded from the study.
Figure 1, replace “µg/l” with “µg/L”. Use SI correctly. Check it through the ms. In addition, <p.001? is that correct? or is p<0.001?
Supplementary
Table S.3
Replace “Total Cases” with “Total cases”
Replace “Refrences” with “References”
Author Response
Comments and Suggestions for Authors
The article needs to be improved and further clarifications are needed before publication.
Material and Methods
What is the study design? It appears to be an analytical observational study, more precisely a cohort design. Follow the STROBE guidelines (https://www.strobe-statement.org/).
Done
Inclusion and exclusion criteria must be clearly defined.
Done
Lanes 134 and 148, replace “ml” with “mL”. Use SI correctly. Check it through the ms.
Done
Results
The characterization of the study population is not evident. The first outcome in an analytical study is to describe the study population.
Add the flowchart showing the women enrolled in the observational study and who were excluded from the study.
Inclusion and exclusion criteria are added
Figure 1, replace “µg/l” with “µg/L”. Use SI correctly. Check it through the ms. In addition, <p.001? is that correct? or is p<0.001?
It is p<0.001 we corrected it
Supplementary
Table S.3
Replace “Total Cases” with “Total cases” Done
Replace “Refrences” with “References” Done
-Line 334: for all of the measured elements!? We rewritten the sentence with correct one
- serum arsenic levels and serum cadmium levels (quartiles)?! From introduction, aim and methods it could not be expected that results of arsenic and cadmium will be provided.
One of additional feature in multivariate is arsenic because in our paper from 2020, we have been able to show that arsenic blood level is strong cancer risk feature for cancers in women from cancer genetic outpatient clinics. Finally in this version we decided to don’t include cadmium blood levels analysis because our pilot studies are indicating that potential interactions between lead and cadmium may be complicated, and they need detailed and more extensive examination. Currently, we cannot say that cadmium itself is a major carcinogenic feature in females.
-Discussion has to be rewritten and improved considering that results of arsenic and cadmium are included.
One of additional feature in multivariate is arsenic because in our paper from 2020, we have been able to show that arsenic blood level is strong cancer risk feature for cancers in women from cancer genetic outpatient clinics. Finally in this version we decided to don’t include cadmium blood levels analysis because our pilot studies are indicating that potential interactions between lead and cadmium may be complicated, and they need detailed and more extensive examination. Currently, we cannot say that cadmium itself is a major carcinogenic feature in females.
- The limitation of this study and conclusion should be improved.
Done
Reviewer 2 Report (Previous Reviewer 3)
Comments and Suggestions for Authors
Dear authors,
I have read your resubmission with ID biomedicines-3651953.
SPECIFIC COMMENTS
- There is still a significant overlap with published literature (up to 33% in Turnitin, excluding references).
- The INTRODUCTION still does not give a background on what heavy metals are before talking about Pb. This could be important to separate carcinogenic heavy metals from non-carcinogenic heavy metals, and explain why lead is of particular concern.
- Very limited discussion of results is presented. At least comparisons should be made to the many studies available for both lead and cadmium levels in humans. With this, the study may generate meaningful discussions and conclusions
KiljaÅ„czyk, A., Matuszczak, M., Marciniak, W., Derkacz, R., Stempa, K., Baszuk, P., BryÅ›kiewicz, M., LubiÅ„ski, K., Cybulski, C., DÄ™bniak, T., Gronwald, J., Huzarski, T., Lener, M. R., Jakubowska, A., Szwiec, M., Stawicka-NieÅ‚acna, M., Godlewski, D., Prusaczyk, A., Jasiewicz, A., Kluz, T., … LubiÅ„ski, J. (2024). Blood Lead Level as Marker of Increased Risk of Ovarian Cancer in BRCA1 Carriers. Nutrients, 16(9), 1370. https://doi.org/10.3390/nu16091370
Gaudet, M. M., Deubler, E. L., Kelly, R. S., Ryan Diver, W., Teras, L. R., Hodge, J. M., Levine, K. E., Haines, L. G., Lundh, T., Lenner, P., Palli, D., Vineis, P., Bergdahl, I. A., Gapstur, S. M., & Kyrtopoulos, S. A. (2019). Blood levels of cadmium and lead in relation to breast cancer risk in three prospective cohorts. International journal of cancer, 144(5), 1010–1016. https://doi.org/10.1002/ijc.31805
Yan et al. (2025) Association of cadmium and lead exposure with mortality in cancer survivors: A prospective cohort study. https://doi.org/10.1016/j.ecoenv.2025.117960
Huang, M., Li, H., Chen, J. et al. Blood lead levels and bladder cancer among US participants: NHANES 1999–2018. BMC Public Health 25, 416 (2025). https://doi.org/10.1186/s12889-025-21549-2
Kim, Y. J., Lee, J. Y., & Seomun, G. (2025). Whole-blood lead, mercury, and cadmium concentrations and their associations with cancer in Korean elders (2007–2018). Archives of Environmental & Occupational Health, 80(1–2), 39–48. https://doi.org/10.1080/19338244.2025.2479107

Author Response
SPECIFIC COMMENTS
- There is still a significant overlap with published literature (up to 33% in Turnitin, excluding references).
We do not want to change text too much because it is after corrections by two English native speakers – co-authors R.Scott and S.Narod
- The INTRODUCTION still does not give a background on what heavy metals are before talking about Pb. This could be important to separate carcinogenic heavy metals from non-carcinogenic heavy metals, and explain why lead is of particular concern.
Done
- Very limited discussion of results is presented. At least comparisons should be made to the many studies available for both lead and cadmium levels in humans. With this, the study may generate meaningful discussions and conclusions
One of additional feature in multivariate is arsenic because in our paper from 2020, we have been able to show that arsenic blood level is strong cancer risk feature for cancers in women from cancer genetic outpatient clinics. Finally in this version we decided to don’t include cadmium blood levels analysis because our pilot studies are indicating that potential interactions between lead and cadmium may be complicated, and they need detailed and more extensive examination. Currently, we cannot say that cadmium itself is a major carcinogenic feature in females. The information about studies concerning lead exposure and cancer risk is presented in supplementary matierals.
Reviewer 3 Report (Previous Reviewer 4)
Comments and Suggestions for Authors
General comments
In compare to the previous version, authors have made some improvements. However, some issues are still present. Again, the basic things are not corrected. Methodological part and results contain lot of Illogically things. For example why arsenic levels and serum cadmium levels were observed for making quartiles and not reported in the title, introduction, abstract, discussion. Authors should again carefully revise manuscript in order to improve the quality. English should be improved.
Major comments
- Again, all the conclusions are still too ambiguous. It is well-known that women over 40 are generally at a higher risk of developing cancer, and the risk is increasing further with age, particularly for breast cancer, colorectal and uterine cancer. In addition, it is well known that spot lead blood levels are indicators of acute exposure. What is going on with cadmium and arsenic?! So complete manuscript has to be rewritten
Special comments
-Having in mind that the obtained data from various studies indicated the association of lead levels with cancer (lung, stomach, bladder, esophagus, brain, and breast), the sentences „High levels of lead are associated with a reduction in cognition. Lead has been judged ty The International Agency for Research on Cancer (IARC) as a probable human carcinogen.” should be deleted and replaced with more proper one in abstract.
-It is strange that authors highlighted the changes of the title although in the previous version the same title was provided. The submitted version with track changes might put reviewers in misunderstanding, which is not acceptable due to the high ethical standards that are expected from authors who submitted research papers. So again, I encourage authors to avoid speculative language. The tittle should be replaced with the more proper one.
-Abstract: The sentence given in Lines 76-77 should be rewritten because it is too speculative.
-In addition, highlights are deleted although mdpi journals do not require highlights.
-Introduction: The first paragraph should be deleted.
-Introduction should be rewritten with the help of English native speaker or professional translator with focus also on cadmium and arsenic.
- Introduction: „Pb paint hazards were noted in households built before 1978 when Pb paint was not generally prohibited. Other sources of the Pb exposure are drinking water service lines made from Pb, Pb solder or plumbing materials that contain Pb [12].” Authors should also highlight the current state. Most of the mentioned was in the past. Please make those things clearer.
-Introduction: “Exposure to Pb from non-occupational sources is also widespread in the adult population, however the levels are generally lower than those found in the workplace.” Concentration ranges in different matrixes should be inserted as well as the references.
-Introduction: In the fourth and fifth paragraph references are missing. It is not acceptable to cite only one article.
- This sentence should be omitted because the study group included women older than 40 years. “Mobilization of Pb from bone to the periphery may also be important in the assessment of toxicity risk of maternal exposure to Pb that may influence the health of any developing foetus and the infants [14].”
-Introduction: Base on the IARC evaluation results inorganic lead compounds are classified into Group 2A while organic lead compounds remain in Group 3. So the sentence “Pb compounds are currently considered by IARC to be probable human carcinogens (group 2B)“ has to be rewritten.
-It is strange to cite Table S.3 as the first one in introduction?!
-Again, the formulation of aim is not proper. The hypothesis is missing.
-How the samples were prepared for ICP-MS analyses. This part is missing in Material and methods.
-Added information about LOQ, LOD are not proper. Please pay more attention to the methodological part because this is the critical step for the research. Information about the monitor isotope and reported isotope for Pb, As and Cd are not provided as well.
-Line 334: for all of the measured elements!?
- serum arsenic levels and serum cadmium levels (quartiles)?! From introduction, aim and methods it could not be expected that results of arsenic and cadmium will be provided.
-Discussion has to be rewritten and improved considering that results of arsenic and cadmium are included.
- The limitation of this study and conclusion should be improved.
Comments on the Quality of English LanguageMinor improvements are required.
Author Response
General comments
In compare to the previous version, authors have made some improvements. However, some issues are still present. Again, the basic things are not corrected. Methodological part and results contain lot of Illogically things. For example why arsenic levels and serum cadmium levels were observed for making quartiles and not reported in the title, introduction, abstract, discussion. Authors should again carefully revise manuscript in order to improve the quality. English should be improved.
Major comments
- Again, all the conclusions are still too ambiguous. It is well-known that women over 40 are generally at a higher risk of developing cancer, and the risk is increasing further with age, particularly for breast cancer, colorectal and uterine cancer. In addition, it is well known that spot lead blood levels are indicators of acute exposure. What is going on with cadmium and arsenic?! So complete manuscript has to be rewritten
According to our data, blood lead level is good marker for chronic exposure except of situation when you are administrating experimentally lead by injection, for cancer risk it is much better to analyze lead blood level instead of lead level in organs which accumulate this element such as, bones liver and brain.
Special comments
-Having in mind that the obtained data from various studies indicated the association of lead levels with cancer (lung, stomach, bladder, esophagus, brain, and breast), the sentences „High levels of lead are associated with a reduction in cognition. Lead has been judged ty The International Agency for Research on Cancer (IARC) as a probable human carcinogen.” should be deleted and replaced with more proper one in abstract.
We rewritten the sentence
-It is strange that authors highlighted the changes of the title although in the previous version the same title was provided. The submitted version with track changes might put reviewers in misunderstanding, which is not acceptable due to the high ethical standards that are expected from authors who submitted research papers. So again, I encourage authors to avoid speculative language. The tittle should be replaced with the more proper one.
We changed title, lead is not potential marker but a marker.
-Abstract: The sentence given in Lines 76-77 should be rewritten because it is too speculative.
This sentence was deleted
-In addition, highlights are deleted although mdpi journals do not require highlights.
-Introduction: The first paragraph should be deleted.
We don’t know what to make because other reviewer asked us to expand this section
-Introduction should be rewritten with the help of English native speaker or professional translator with focus also on cadmium and arsenic.
Manuscript was edited by two English native speakers co-authors of paper S. Narod and R. Scott. One of additional feature in multivariate is arsenic because in our paper from 2020, we have been able to show that arsenic blood level is strong cancer risk feature for cancers in women from cancer genetic outpatient clinics. Finally in this version we decided to don’t include cadmium blood levels analysis because our pilot studies are indicating that potential interactions between lead and cadmium may be complicated, and they need detailed and more extensive examination. Currently, we cannot say that cadmium itself is a major carcinogenic feature in females.
- Introduction: „Pb paint hazards were noted in households built before 1978 when Pb paint was not generally prohibited. Other sources of the Pb exposure are drinking water service lines made from Pb, Pb solder or plumbing materials that contain Pb [12].” Authors should also highlight the current state. Most of the mentioned was in the past. Please make those things clearer.
Currently it is difficult to indicate special non-occupational exposure probably it is airplane gas and from our experience we found Pb enrichement in some fruits products.
-Introduction: “Exposure to Pb from non-occupational sources is also widespread in the adult population, however the levels are generally lower than those found in the workplace.” Concentration ranges in different matrixes should be inserted as well as the references.
Currently it is difficult to indicate special non-occupational exposure probably it is airplane gas and from our experience we found Pb enrichement in some fruits products.
-Introduction: In the fourth and fifth paragraph references are missing. It is not acceptable to cite only one article.
We added additional studies.
- This sentence should be omitted because the study group included women older than 40 years. “Mobilization of Pb from bone to the periphery may also be important in the assessment of toxicity risk of maternal exposure to Pb that may influence the health of any developing foetus and the infants [14].”
We omitted this sentence
-Introduction: Base on the IARC evaluation results inorganic lead compounds are classified into Group 2A while organic lead compounds remain in Group 3. So the sentence “Pb compounds are currently considered by IARC to be probable human carcinogens (group 2B)“ has to be rewritten.
The sentence is rewritten.
-It is strange to cite Table S.3 as the first one in introduction?!
We agree with reviewer but this table is large this is why we put it into supplementary materials.
-Again, the formulation of aim is not proper. The hypothesis is missing.
Done
-How the samples were prepared for ICP-MS analyses. This part is missing in Material and methods.
Done
-Added information about LOQ, LOD are not proper. Please pay more attention to the methodological part because this is the critical step for the research. Information about the monitor isotope and reported isotope for Pb, As and Cd are not provided as well.
Done
-Line 334: for all of the measured elements!?
Sentence was rewritten.
- serum arsenic levels and serum cadmium levels (quartiles)?! From introduction, aim and methods it could not be expected that results of arsenic and cadmium will be provided.
One of additional feature in multivariate is arsenic because in our paper from 2020, we have been able to show that arsenic blood level is strong cancer risk feature for cancers in women from cancer genetic outpatient clinics. Finally in this version we decided to don’t include cadmium blood levels analysis because our pilot studies are indicating that potential interactions between lead and cadmium may be complicated, and they need detailed and more extensive examination. Currently, we cannot say that cadmium itself is a major carcinogenic feature in females.
-Discussion has to be rewritten and improved considering that results of arsenic and cadmium are included.
- The limitation of this study and conclusion should be improved.
Done
Round 2
Reviewer 1 Report (Previous Reviewer 2)
Comments and Suggestions for Authors
The authors did a good job responding to my previous concerns.
Minor comments:
Abstract, replace "methodes" with "methods"
Author Response
Abstract, replace "methodes" with "methods"
Done
Reviewer 2 Report (Previous Reviewer 3)
Comments and Suggestions for Authors
The authors gave responses but it is now impossible to assess if any changes were made. They seem to copy and paste the initial version of the manuscript upwards in an unformatted manner this time.
As for plagiarism, the authors cite that they do not want to change text too much because it is after corrections by two English native speakers. Yet the plagiarism percentage seems high, and this I leeave to the Editorial office to decide
Author Response
The authors gave responses but it is now impossible to assess if any changes were made. They seem to copy and paste the initial version of the manuscript upwards in an unformatted manner this time.
As for plagiarism, the authors cite that they do not want to change text too much because it is after corrections by two English native speakers. Yet the plagiarism percentage seems high, and this I leave to the Editorial office to decide
Reviewer 3 Report (Previous Reviewer 4)
Comments and Suggestions for Authors
In compared with the previous version, authors have made some improvements. Some issues are still present.
-Title should be as “Blood lead (Pb) levels as a possible marker of cancer risk in a prospective cohort of women with non-occupational exposure”
-Abstract, sentence given in lines 55-56 should be deleted.
-Introduction, line 104: sentence should be rewritten.
- Introduction, in lines 118-119: “Blood Pb levels can enter the body by inhaling, swallowing or by moving from deeper compartments of the body.” The sentence should be rewritten.
-Sentence given in line 119 should be deleted: “The level of Pb in the bloodstream is proportional to the amount of Pb found in other tissues, including the kidneys.”
-Sentence given in line 140 should be deleted “Pb compounds are currently considered by IARC to be a probable human carcinogens (group 2B)” .
-Again, authors did not reorder the numbers of tables in supplement material. It is strange to firstly cite Table S.3 in introduction.
-The revised aim and the given hypothesize should be improved. English is not proper. Please see comments in previous revision steps.
- Having in mind that authors decided to include data for arsenic in the same samples it should be underlined why this is important in the introduction section (aim/hypothesis).
-The methodological part should be improved. It is not common to analyze lead and arsenic directly in the blood sample without previous digestion. Please explain in detail the sample preparation steps.
-Quality control: First sentence should be rewritten. Please improve this part with the help of professionals in English.
- LOQ and LOD values for arsenic are missing.
-Information about the monitor isotope and reported isotope are not provided as well.
-Table 5 and Table 6 are not mentioned in the text in section Results.
-Text given in lines 354-355 should be omitted “suggesting there may be tolerance to the effects of Pb ingested and stored over longer periods of time compared more recent exposures.”
-Text given in lines 356-362 should be rewritten with the help of professionals in English.
-Please rewrite the whole discussion section. The discussion section is part where the author connects their findings to the existing literature and provides a more in-depth explanation of the research's implications. At this stage discussion is not fluent and should be improved.
-Please give more efforts in identifying the limitations.
-Conclusion is not proper.
Comments on the Quality of English LanguageEnglish should be improved especially in the discussion section.
Author Response
In compared with the previous version, authors have made some improvements. Some issues are still present.
-Title should be as “Blood lead (Pb) levels as a possible marker of cancer risk in a prospective cohort of women with non-occupational exposure”
Done
-Abstract, sentence given in lines 55-56 should be deleted.
Done
-Introduction, line 104: sentence should be rewritten.
Done
- Introduction, in lines 118-119: “Blood Pb levels can enter the body by inhaling, swallowing or by moving from deeper compartments of the body.” The sentence should be rewritten.
Done
-Sentence given in line 119 should be deleted: “The level of Pb in the bloodstream is proportional to the amount of Pb found in other tissues, including the kidneys.”
Done
-Sentence given in line 140 should be deleted “Pb compounds are currently considered by IARC to be a probable human carcinogens (group 2B)” .
Done
-Again, authors did not reorder the numbers of tables in supplement material. It is strange to firstly cite Table S.3 in introduction.
Done
-The revised aim and the given hypothesize should be improved. English is not proper. Please see comments in previous revision steps.
Done
- Having in mind that authors decided to include data for arsenic in the same samples it should be underlined why this is important in the introduction section (aim/hypothesis).
Done
-The methodological part should be improved. It is not common to analyze lead and arsenic directly in the blood sample without previous digestion. Please explain in detail the sample preparation steps.
There is no need to digest samples. Despite the fact that blood is a complex matrix there is a lot of evidence that it is possible to determine metals directly in blood. We added more detailed section in methods, as recommended.
-Quality control: First sentence should be rewritten. Please improve this part with the help of professionals in English.
Done
- LOQ and LOD values for arsenic are missing.
Done
-Information about the monitor isotope and reported isotope are not provided as well.
It was mentioned: “The determination of 208Pb and 75As was carried out using the ICP-MS mass spectrometer ELAN DRC-e. Oxygen was used as a reaction gas. Arsenic was measured in mass-shift mode at mass 91 as 91AsO+.”
-Table 5 and Table 6 are not mentioned in the text in section Results.
Done
-Text given in lines 354-355 should be omitted “suggesting there may be tolerance to the effects of Pb ingested and stored over longer periods of time compared more recent exposures.”
The sentence was rewritten.
-Text given in lines 356-362 should be rewritten with the help of professionals in English.
-Please rewrite the whole discussion section. The discussion section is part where the author connects their findings to the existing literature and provides a more in-depth explanation of the research's implications. At this stage discussion is not fluent and should be improved.
We rewrote the discussion.
-Please give more efforts in identifying the limitations.
Done
-Conclusion is not proper.
Done
Round 3
Reviewer 3 Report (Previous Reviewer 4)
Comments and Suggestions for Authors
Unfortunately, authors did not address some issues although they wrote „done“ in replies. Please take below comments into consideration with the help of an expert in English.
-Again, authors should think about changing the title. Title should be as “Blood lead (Pb) levels as a possible marker of cancer risk in a prospective cohort of women with non-occupational exposure”
-Abstract, was much better in the previous version. The text given in the background should be improved with the help of professionals in English.
-Abstract, line 50-51. The sentence “This study suggests that elevated blood Pb levels could be used as a marker of cancer risk in women who are not occupationally exposed to this element.” should be omitted in the subheading Results in Abstract.
-Abstract, line 58-59. The sentence “In summary, exposure to Pb may result in an increased cancer risk in women who are not occupationally exposed to this element.” should be omitted in the subheading Conclusion in Abstract.
-Introduction, line 93: delete the text “, which means it remains unclassifiable”.
- Introduction, sentence in lines 113-115: should be rewritten.
-Introduction, the text given in lines 149-153 and 139-140 should be combined and rewritten as “The current study was undertaken to better define the relationship between Pb exposure and cancer risk. To address the shortfall in knowledge about the relationship between Pb and cancer risk, especially in Poland, we undertook a study using a large cohort of unaffected women where we measured baseline Pb levels and documented incident cancers for an average of 6 years follow-up. We included the blood arsenic level in multivariate analysis due to the fact that blood arsenic(As) level has been recognized recently as additional significant marker of cancer risk in women [29].”
-2.2. Measurement of blood Pb, sentence in line 174 should not begin with the number (10 ml). It should be rewritten.
-Statistical analysis, sentence given in lines 218-219 should be divided into two. Pay attention on English!
-Results, line 263: “1.46” should be outside the brackets.
-Discussion, line 347: reference is missing.
-Discussion, line 348: instead “By including” it should be “By comparing”. The whole sentence is too long. Please divide it into two.
-Discussion, line 352-353: The sentence should be rewritten.
- Discussion, line 358-359: The sentence should be rewritten.
- Discussion, line 366-368: The sentence is too long and should be rewritten.
-Discussion, line 369-371: The sentence should be rewritten as “Given that young women with higher blood Pb levels had an almost 3-fold increased risk of developing breast cancer compared to women with the lowest levels, the cut-off values of 9.39 µg /l should be further explored”.
-Text given in the limitation in lines 378-382 should be deleted.
-Conclusion, sentence given in line 389 should be deleted.
Comments on the Quality of English LanguageThe improvements are required in some parts.
Author Response
Again, authors should think about changing the title. Title should be as “Blood lead (Pb) levels as a possible marker of cancer risk in a prospective cohort of women with non-occupational exposure”
-Abstract, was much better in the previous version. The text given in the background should be improved with the help of professionals in English.
Done
-Abstract, line 50-51. The sentence “This study suggests that elevated blood Pb levels could be used as a marker of cancer risk in women who are not occupationally exposed to this element.” should be omitted in the subheading Results in Abstract.
Done
-Abstract, line 58-59. The sentence “In summary, exposure to Pb may result in an increased cancer risk in women who are not occupationally exposed to this element.” should be omitted in the subheading Conclusion in Abstract.
Done
-Introduction, line 93: delete the text “, which means it remains unclassifiable”.
Done
- Introduction, sentence in lines 113-115: should be rewritten.
Done
-Introduction, the text given in lines 149-153 and 139-140 should be combined and rewritten as “The current study was undertaken to better define the relationship between Pb exposure and cancer risk. To address the shortfall in knowledge about the relationship between Pb and cancer risk, especially in Poland, we undertook a study using a large cohort of unaffected women where we measured baseline Pb levels and documented incident cancers for an average of 6 years follow-up. We included the blood arsenic level in multivariate analysis due to the fact that blood arsenic(As) level has been recognized recently as additional significant marker of cancer risk in women [29].”
Done
-2.2. Measurement of blood Pb, sentence in line 174 should not begin with the number (10 ml). It should be rewritten.
Done
-Statistical analysis, sentence given in lines 218-219 should be divided into two. Pay attention on English!
Done
-Results, line 263: “1.46” should be outside the brackets.
Done
-Discussion, line 347: reference is missing.
Reference is not missing it is number 31
-Discussion, line 348: instead “By including” it should be “By comparing”. The whole sentence is too long. Please divide it into two.
Done
-Discussion, line 352-353: The sentence should be rewritten.
Done
- Discussion, line 358-359: The sentence should be rewritten.
Done
- Discussion, line 366-368: The sentence is too long and should be rewritten.
Done
-Discussion, line 369-371: The sentence should be rewritten as “Given that young women with higher blood Pb levels had an almost 3-fold increased risk of developing breast cancer compared to women with the lowest levels, the cut-off values of 9.39 µg /l should be further explored”.
Done
-Text given in the limitation in lines 378-382 should be deleted.
Done
-Conclusion, sentence given in line 389 should be deleted.
Done
This manuscript is a resubmission of an earlier submission. The following is a list of the peer review reports and author responses from that submission.
Round 1
Reviewer 1 Report
Comments and Suggestions for Authors
This is a very interesting article.
The following changes should be made. Major changes
INTRODUCTION
1) The introduction should mention that lead was used as an antiknock agent in gasoline and has been progressively disappearing.
2) We should also talk not only about cancer but also about the effects of lead on health in general. Lead behaves like calcium and accumulates in the bones. It alters all the processes in which calcium is involved, producing neurological disorders, hematological problems, etc.
3) Lead accumulated in the bones can be mobilized in women during pregnancy and menopause when calcium is mobilized.
4) Exposure to lead and other heavy metals from fish consumption, such as tuna, should also be discussed.
5) In addition, there may be environmental sources in old houses in cities where the plumbing in old houses may be lead. This increases social inequalities since the most disadvantaged people may be more exposed to lead.
6) There may be environmental sources of lead exposure, such as the grounds of old mines or smelters, where lead is present..
7) We should also talk about how the lead enters the organism, e.g. via oral, respiratory etc.
8) Sugerimos que los autores incluyan un DAG (gráfico acíclico dirigido) en la introducción, para explicar la relación entre exposición y cáncer. Hay muchos programas gratuitos como dagitty que pueden hacerlo.
MATERIAL AND METHODS
8) There is No control for socioeconomic status (SES), and environmental factors which could influence both lead exposure and cancer risk. The authors should adjust the model alt least by SES. If adjustment by environmental factors is not possible , it should be commented on the discussion as a limitation.
9) The authors should analyze if there is potential collinearity problems (see point 12 below
RESULTs
10) In Table 3, the reported HR=1.34 (95% CI: 0.91-1.98; p=0.13) for any cancer when comparing the three highest quartiles to the lowest is not significant (p>0.05). However, the article suggests a causal relationship without robust justification.
11) The association for women under 50 years old (HR=2.6; 95% CI: 1.38-4.91; p=0.003) is significant, but it is unclear if this subgroup was pre-specified or emerged from a post-hoc analysis, raising concerns about data dredging. The authors should explain why this cut-off point was set. What biological or scientific justification exists?
12) The authors didn’t do any analysis of other heavy metal exposures (e.g., arsenic, cadmium), which are known to have carcinogenic effects. If they hav information they should introduce it in the models, other wise they should comment in the limitation section of the discussion.
13) In Table 3, the hazard ratio (HR) for cancer in the highest quartile (Q4) among women under 50 is lower (HR=2.1, p=0.08) than in quartile 2 (HR=2.5, p=0.01). This is unusual and suggests categorization errors or issues in model adjustments.
DISCUSSION
14) The authors should comment in the discussion as a limitation that they uses a single blood lead measurement to assess long-term exposure. Lead has a half-life of 4-5 weeks in blood, meaning this measurement does not reflect chronic exposure. A more accurate approach would involve bone lead biomarkers or multiple blood measurements over time.
15) Comment on the results of point 11 above. A review of potential Are there any collinearity problems or lead misclassification?
16) The discussion claims the results “demonstrate a carcinogenic effect of lead”, even though most HRs are not statistically significant (p>0.05).
17) The study states that lead exposure increases cancer risk, yet the study design does not establish causality.
18) One of the major inconsistencies in the article is the contradictory interpretation of the relationship between blood lead levels and cancer risk. In different sections, the study presents conflicting conclusions, suggesting that low lead levels are protective and higher lead levels are associated with increased cancer risk. These inconsistencies need to be addressed for clarity and scientific rigor.
The core contradiction is that the study cannot simultaneously argue that high lead levels increase cancer risk and that low lead levels protect against cancer—they are two sides of the same relationship.
a. Claim that Low Blood Lead Levels Reduce Cancer Risk
Lines 168-174 " This paragraph implies that lower lead levels may be protective against cancer, particularly in women under 50. However, the hazard ratio (HR=1.34; p=0.13) for the general population is not statistically significant, and the increased risk in younger women should not be interpreted as evidence that low lead levels are protective
B= Results Suggest High Lead Increases Cancer Risk
Location: Results, Table 3 and Table 4 "Women with lead levels above 9.39 µg/L had an increased risk of developing any cancer. The hazard ratio for women in quartiles 2-4 combined compared to quartile 1 was HR=1.34; (95% CI: 0.91-1.98; p=0.13). Among the subgroup of women below 50 years of age, the observed effect was stronger for any cancer (Q1 vs Q2-4; HR=2.6; (95% CI: 1.38-4.91; p=0.003)."
C= Discussion Implies No Effect in Older Women
Location: Discussion, Lines 175-177 Excerpt from the article: "For postmenopausal women there appeared to be no association with cancer risk and Pb levels. Of interest was the finding that overall, women over 50 years of age generally had higher levels of Pb compared to women younger than 50, suggesting there may be tolerance to the effects of Pb ingested over longer periods of time as observed in older women." This paragraph suggests that higher lead levels do not increase cancer risk in postmenopausal women, contradicting the claim that high lead levels are associated with cancer risk.
Author Response
Reviewer number 1
INTRODUCTION
1) The introduction should mention that lead was used as an antiknock agent in gasoline and has been progressively disappearing.
-Done
2) We should also talk not only about cancer but also about the effects of lead on health in general. Lead behaves like calcium and accumulates in the bones. It alters all the processes in which calcium is involved, producing neurological disorders, hematological problems, etc.
-Done
3) Lead accumulated in the bones can be mobilized in women during pregnancy and menopause when calcium is mobilized.
-Done
4) Exposure to lead and other heavy metals from fish consumption, such as tuna, should also be discussed.
-Done
5) In addition, there may be environmental sources in old houses in cities where the plumbing in old houses may be lead. This increases social inequalities since the most disadvantaged people may be more exposed to lead.
-Done
6) There may be environmental sources of lead exposure, such as the grounds of old mines or smelters, where lead is present.
-Done
7) We should also talk about how the lead enters the organism, e.g. via oral, respiratory etc.
-Done
MATERIAL AND METHODS
8) There is No control for socioeconomic status (SES), and environmental factors which could influence both lead exposure and cancer risk. The authors should adjust the model alt least by SES. If adjustment by environmental factors is not possible , it should be commented on the discussion as a limitation.
We added comment that an SES status was not studied
9) The authors should analyze if there is potential collinearity problems (see point 12 below
RESULTS
We added reults from cadmium and arsenic level
10) In Table 3, the reported HR=1.34 (95% CI: 0.91-1.98; p=0.13) for any cancer when comparing the three highest quartiles to the lowest is not significant (p>0.05). However, the article suggests a causal relationship without robust justification.
After adding results on cadmium and arsenic levels in multivariate COX regression HR become statistically significant
11) The association for women under 50 years old (HR=2.6; 95% CI: 1.38-4.91; p=0.003) is significant, but it is unclear if this subgroup was pre-specified or emerged from a post-hoc analysis, raising concerns about data dredging. The authors should explain why this cut-off point was set. What biological or scientific justification exists?
Generally in all analysis we are doing on groups of adult females we are checking differences between subgroups below and above 50 years of age, the rational for this is that age of 50 is connected with menopause status.
12) The authors didn’t do any analysis of other heavy metal exposures (e.g., arsenic, cadmium), which are known to have carcinogenic effects. If they hav information they should introduce it in the models, other wise they should comment in the limitation section of the discussion.
We introduce in the model results about cadmium and arsenic quartiles levels.
13) In Table 3, the hazard ratio (HR) for cancer in the highest quartile (Q4) among women under 50 is lower (HR=2.1, p=0.08) than in quartile 2 (HR=2.5, p=0.01). This is unusual and suggests categorization errors or issues in model adjustments.
According to our experience situations such as in Pb studies we observe at least a few times studying correlations between elements and cancer risk. After additional validation this were never artefacts.
DISCUSSION
14) The authors should comment in the discussion as a limitation that they uses a single blood lead measurement to assess long-term exposure. Lead has a half-life of 4-5 weeks in blood, meaning this measurement does not reflect chronic exposure. A more accurate approach would involve bone lead biomarkers or multiple blood measurements over time.
We added in limitation section the use of single blood lead measurement
15) Comment on the results of point 11 above. A review of potential Are there any collinearity problems or lead misclassification?
As we mentioned above we did not find collinearity problem
16) The discussion claims the results “demonstrate a carcinogenic effect of lead”, even though most HRs are not statistically significant (p>0.05).
As we mentioned above date become statistically significant
17) The study states that lead exposure increases cancer risk, yet the study design does not establish causality.
We introduce statements that our studies did not analyze Pb causality
18) One of the major inconsistencies in the article is the contradictory interpretation of the relationship between blood lead levels and cancer risk. In different sections, the study presents conflicting conclusions, suggesting that low lead levels are protective and higher lead levels are associated with increased cancer risk. These inconsistencies need to be addressed for clarity and scientific rigor.
Statements that low blood lead level is protective against cancer risk was removed.
The core contradiction is that the study cannot simultaneously argue that high lead levels increase cancer risk and that low lead levels protect against cancer—they are two sides of the same relationship.
a. Claim that Low Blood Lead Levels Reduce Cancer Risk
Lines 168-174 " This paragraph implies that lower lead levels may be protective against cancer, particularly in women under 50. However, the hazard ratio (HR=1.34; p=0.13) for the general population is not statistically significant, and the increased risk in younger women should not be interpreted as evidence that low lead levels are protective
B= Results Suggest High Lead Increases Cancer Risk
Location: Results, Table 3 and Table 4 "Women with lead levels above 9.39 µg/L had an increased risk of developing any cancer. The hazard ratio for women in quartiles 2-4 combined compared to quartile 1 was HR=1.34; (95% CI: 0.91-1.98; p=0.13). Among the subgroup of women below 50 years of age, the observed effect was stronger for any cancer (Q1 vs Q2-4; HR=2.6; (95% CI: 1.38-4.91; p=0.003)."
We made statements that according to our date Pb is just marker of cancer risk
C= Discussion Implies No Effect in Older Women
Location: Discussion, Lines 175-177 Excerpt from the article: "For postmenopausal women there appeared to be no association with cancer risk and Pb levels. Of interest was the finding that overall, women over 50 years of age generally had higher levels of Pb compared to women younger than 50, suggesting there may be tolerance to the effects of Pb ingested over longer periods of time as observed in older women." This paragraph suggests that higher lead levels do not increase cancer risk in postmenopausal women, contradicting the claim that high lead levels are associated with cancer risk.
The above problem was discussed with suggestion that estradiol (not estrons) is critical for carcinogenetic effect of Pb.
Reviewer 2 Report
Comments and Suggestions for Authors
The authors present what appears to be a cohort study design in which their findings suggest that lead exposure contributes as a risk factor for the development of cancer in the female population. I have several concerns aiming at enhancing the interest of the paper and at giving more precisions.
- Lane 56, replace "Based on meta-analysis released in 2024[21] 11 studies" with "Based on meta-analysis released in 2024 [21], 11 studies"
- Lane 69, replace "breast disease[Table S5]" with "breast disease [Table S5]"
- Methods, add the study design. It appears to be an observational cohort study. Follow the STROBE guidelines for cohort studies https://www.strobe-statement.org/.
- Lane 85, replace “ml” with 2mL”. The international system of units (SI) should be used. Check it through the manuscript.
- Statistical analysis, were all the data normally distributed? If not, what statistical tests were used?
- For better understanding, the format of Table 1 should be corrected. Use punctuation marks appropriately.
- Use percentages in Table 2 to visualize better the number of subjects with cancer and each of its types.
Author Response
Reviewer number 2
The authors present what appears to be a cohort study design in which their findings suggest that lead exposure contributes as a risk factor for the development of cancer in the female population. I have several concerns aiming at enhancing the interest of the paper and at giving more precisions.
- Lane 56, replace "Based on meta-analysis released in 2024[21] 11 studies" with "Based on meta-analysis released in 2024 [21], 11 studies"
-Done.
- Lane 69, replace "breast disease[Table S5]" with "breast disease [Table S5]"
-Done.
- Methods, add the study design. It appears to be an observational cohort study. Follow the STROBE guidelines for cohort studies https://www.strobe-statement.org/.
-Done in section study group.
- Lane 85, replace “ml” with 2mL”. The international system of units (SI) should be used. Check it through the manuscript.
-Done.
- Statistical analysis, were all the data normally distributed? If not, what statistical tests were used?
In order to determine the relationship between independent variables on time to event, Cox proportional hazard models were used.
All variables included in the model are qualitative variables so checking for normality of distribution is not applicable in this case.
- For better understanding, the format of Table 1 should be corrected. Use punctuation marks appropriately.
-Done.
- Use percentages in Table 2 to visualize better the number of subjects with cancer and each of its types.
-Done.
Reviewer 3 Report
Comments and Suggestions for Authors
Dear authors,
I have now read your submission with ID biomedicines-3529187. The overall impression is that this study is a nice contribution to the body of literature. However, the authors transferred this submission from another journal, and spared little effort to format it to the journal style, which overall makes it hard to read. For example, Table 4 is scattered, and not very readable in the current state. Please consider the following suggestions and comments to improve the submission.
- The INTRODUCTION should first give a background on what heavy metals are before talking about Pb.
- Please be consistent with the use of the words lead levels, Pb levels and blood lead.
- Methods
- The ICP-MS method analyzed a single blood sample per subject. Given lead’s short biological half-life (~30 days), how does this measurement reflect long-term exposure?
- The 41-month delay between blood collection and analysis introduces potential degradation concerns. Were lead stability tests conducted?
- The study does not clarify if batch effects were controlled when running ICP-MS on different sets of samples.
- The authors stated that technicians were blinded (see L209) but I was wondering if this was for all the processing steps?
- In Tables 1 and 2, please include standard deviations with the mean Pb levels.
- In Tables 3 & 4, were hazard ratios adjusted for multiple testing correction (for example, Bonferroni correction)?
- Visualization of lead quartiles using Kaplan-Meier curves could better analyze time-to-event data (i.e., the time until the cancer occurs).
- L156: It is stated that there is a statistically significant association between blood lead levels and cancer risk, but some p-values are not significant (p=0.13 for the general population).
- The results for women under 50 (HR=2.6, p=0.003) is interesting but the authors do not clearly justify why younger women would be more susceptible. Could there be confounding factors such as occupational exposure, dietary habits, or hormonal influences?
- The Cox proportional hazards model is used, but it is unclear if proportional hazard assumptions were checked. Was a Schoenfeld residual test performed?
- The hazard ratios for different quartiles (Tables 3 & 4) show inconsistent trends (e.g., Q3 having a higher HR than Q4 in some cases). Was there an issue with sample distribution that have not been mentioned by the authors?
- Several questions should be asked about confounding variables:
- Lifestyle factors (such as diet, alcohol consumption, environmental lead exposure beyond blood levels) are nowhere accounted for in the present study?
- The exclusion of BRCA1 mutation carriers is appropriate, but were other genetic factors considered?
- Smoking status is included in the multivariable model, but how was pack-year exposure handled? Could passive smoking be an unrecognized contributing factor?
- Other comments are in the attached manuscript PDF.
- This manuscript also has substantial overlap with published literature records (upto 45%) with the greatest percentage from https://onlinelibrary.wiley.com/doi/full/10.1002/ijc.32595

I strongly recommend that the manuscript is thoroughly reviewed by a proficient English-speaking scientist
Author Response
Reviewer number 3
- The INTRODUCTION should first give a background on what heavy metals are before talking about Pb.
-Done.
- Please be consistent with the use of the words lead levels, Pb levels and blood lead.
-Done.
- Methods
- The ICP-MS method analyzed a single blood sample per subject. Given lead’s short biological half-life (~30 days), how does this measurement reflect long-term exposure?
According to our experience in population without special exposures we are not expecting such changes within around 6 years.
- The 41-month delay between blood collection and analysis introduces potential degradation concerns. Were lead stability tests conducted?
-Done.
- The study does not clarify if batch effects were controlled when running ICP-MS on different sets of samples.
-Done.
- The authors stated that technicians were blinded (see L209) but I was wondering if this was for all the processing steps?
-Done.
- In Tables 1 and 2, please include standard deviations with the mean Pb levels.
-Done.
- In Tables 3 & 4, were hazard ratios adjusted for multiple testing correction (for example, Bonferroni correction)?
By using mulivariate Cox regression Bonferroni correction is not requaired
- Visualization of lead quartiles using Kaplan-Meier curves could better analyze time-to-event data (i.e., the time until the cancer occurs).
-Done.
- L156: It is stated that there is a statistically significant association between blood lead levels and cancer risk, but some p-values are not significant (p=0.13 for the general population).
After adding arsenic and cadmium quartile levels data become significant.
- The results for women under 50 (HR=2.6, p=0.003) is interesting but the authors do not clearly justify why younger women would be more susceptible. Could there be confounding factors such as occupational exposure, dietary habits, or hormonal influences?
-Done.
- The Cox proportional hazards model is used, but it is unclear if proportional hazard assumptions were checked. Was a Schoenfeld residual test performed?
Cox Proportional Hazard models used for calculations were tested for compliance with the proportional hazards assumption. The presented models met this criterion both in the global and specific context of each variable included in the model,
except for the variable describing smoking in the subgroup containing only patients with breast cancer in patients under 50 years of age (p = 0.016).
-
- The hazard ratios for different quartiles (Tables 3 & 4) show inconsistent trends (e.g., Q3 having a higher HR than Q4 in some cases). Was there an issue with sample distribution that have not been mentioned by the authors?
For us such trends we could see in few times in correlation between elements and cancer risk and it where not artefacts.
- Several questions should be asked about confounding variables:
- Lifestyle factors (such as diet, alcohol consumption, environmental lead exposure beyond blood levels) are nowhere accounted for in the present study?
We plan to study the above lifestyle factors in the future.
- The exclusion of BRCA1 mutation carriers is appropriate, but were other genetic factors considered?
Our genetic factors were not considered in this paper but we plan to study in the future
- Smoking status is included in the multivariable model, but how was pack-year exposure handled? Could passive smoking be an unrecognized contributing factor?
We didn’t check the pack-year exposure, for majority of women smoking is not a major factor of the cancer risk
- Other comments are in the attached manuscript PDF.
-Done
- This manuscript also has substantial overlap with published literature records (upto 45%) with the greatest percentage from https://onlinelibrary.wiley.com/doi/full/10.1002/ijc.32595
-Done
Reviewer 4 Report
Comments and Suggestions for Authors
General comments
Lead is one of the nonessential elements. Base on the IARC evaluation results inorganic lead compounds are classified into Group 2A while organic lead compounds remain in Group 3. Possible mechanisms of lead toxicity include: DNA damage by reactive oxygen species; the disruption of DNA synthesis and repair as well as the interference in cell cycle control; the impairment of the expression of cancer-related genes. The obtained data from various studies indicated the association of lead levels with cancer (lung, stomach, bladder, esophagus, brain, and breast). Hence, the obtained data in this cohort might be useful. However, authors conclude “We found a statistically significant association between a low blood lead levels and the subsequent risk of cancer in women.” Having in mind that the results is based on just one spot blood test of lead, all the conclusions are too ambiguous. In addition, it is well known that lead blood levels are indicators of acute exposure and that lead has tendency to bioaccumulate. The obtained results indicate that other risk factors might be more pronounced. It is very important also to state what happens with women without lead in blood. In addition, the aim of the study is not clear. Authors should carefully revise manuscript in order to improve the quality. Some additional issues have to be addressed.
Special comments
-Abstract: Please rewrite the sentence give in lines 30-31 as well as in the introduction in lines 53-54.
- Impersonal language should be used in the whole text. It is characterised by the avoidance of personal pronouns.
-Line 35: correct font and include if microwave digestion was used.
-Line 36: 6 or 16 or both?!
-Line 39-41: Please rewrite both sentences. There are lot of associated risk factors that might also contribute to onset of cancer that were not discussed. So, the obtained results could be taken with preclusion.
-Text give in lines 56-69 has to be carefully rewritten. Please cite references and carefully check recently published studies (see https://www.mdpi.com/2305-6304/12/7/490)
- Female gender is the strongest breast cancer risk factor. So please avoid speculative language. The tittle does not correspond to the text given in the introduction “Current study was undertaking to
correctly define the relationship between lead exposure and breast cancer risk. Since is the 67
most common malignancy in Poland we specificity focus on the relationship between lead 68
levels and breast disease.”
-line 69-Table S5-please correct numbers
-The aim of the study as well as the hypothesis should be clearly formulated.
-Introduction section does not contain any explanation of possible mechanism of lead induced carcinogenesis.
-Study group: please better explain inclusion and exclusion criteria.
-Sample preparation, ICP-MS instrument condition together with quantification are missing (LOQ, LOD, …).
-Please format all tables according to the journal guidelines
-Follow up period is not clear enough. Please specify better in material and methods section.
-Line 170-171: Please carefully check the sentence. Based on the p-value this is not significant. HR should be much above 1.
-Discussion has to be rewritten and improved.
-Special attention should be paid to the limitation of this study.
-Conslusion should be expanded. Please avoid spacualtive language.
Author Response
Reviewer number 4
Lead is one of the nonessential elements. Base on the IARC evaluation results inorganic lead compounds are classified into Group 2A while organic lead compounds remain in Group 3. Possible mechanisms of lead toxicity include: DNA damage by reactive oxygen species; the disruption of DNA synthesis and repair as well as the interference in cell cycle control; the impairment of the expression of cancer-related genes. The obtained data from various studies indicated the association of lead levels with cancer (lung, stomach, bladder, esophagus, brain, and breast). Hence, the obtained data in this cohort might be useful. However, authors conclude “We found a statistically significant association between a low blood lead levels and the subsequent risk of cancer in women.” Having in mind that the results is based on just one spot blood test of lead, all the conclusions are too ambiguous. In addition, it is well known that lead blood levels are indicators of acute exposure and that lead has tendency to bioaccumulate. The obtained results indicate that other risk factors might be more pronounced. It is very important also to state what happens with women without lead in blood. In addition, the aim of the study is not clear. Authors should carefully revise manuscript in order to improve the quality. Some additional issues have to be addressed.
Special comments
-Abstract: Please rewrite the sentence give in lines 30-31 as well as in the introduction in lines 53-54.
-Done.
- Impersonal language should be used in the whole text. It is characterised by the avoidance of personal pronouns.
-Done.
-Line 35: correct font and include if microwave digestion was used
-Done.
-Line 36: 6 or 16 or both?!
-It means 6years and 2months(6,16years)
-Line 39-41: Please rewrite both sentences. There are lot of associated risk factors that might also contribute to onset of cancer that were not discussed. So, the obtained results could be taken with preclusion.
Done.
-Text give in lines 56-69 has to be carefully rewritten. Please cite references and carefully check recently published studies (see https://www.mdpi.com/2305-6304/12/7/490)
-Done.
- Female gender is the strongest breast cancer risk factor. So please avoid speculative language. The tittle does not correspond to the text given in the introduction “Current study was undertaking to
-Done.
correctly define the relationship between lead exposure and breast cancer risk. Since is the 67
-Done.
most common malignancy in Poland we specificity focus on the relationship between lead 68
levels and breast disease.”
-Done.
-line 69-Table S5-please correct numbers
-Done
-The aim of the study as well as the hypothesis should be clearly formulated.
-Done
-Introduction section does not contain any explanation of possible mechanism of lead induced carcinogenesis.
-Done
-Study group: please better explain inclusion and exclusion criteria.
-Done
-Sample preparation, ICP-MS instrument condition together with quantification are missing (LOQ, LOD, …).
-Done
-Please format all tables according to the journal guidelines
-Done
-Follow up period is not clear enough. Please specify better in material and methods section.
-Done
-Line 170-171: Please carefully check the sentence. Based on the p-value this is not significant. HR should be much above 1.
-Discussion has to be rewritten and improved.
-Done
-Special attention should be paid to the limitation of this study.
-Done
-Conslusion should be expanded. Please avoid spacualtive language.
-Done